# SLAF-seq Uncovers the Genetic Diversity and Adaptation of Chinese Elm (*Ulmus parvifolia*) in Eastern China

**Yun-zhou Lyu** **, Xiao-yun Dong, Li-bin Huang, Ji-wei Zheng, Xu-dong He, Hai-nan Sun and Ze-ping Jiang \***

Jiangsu Academy of Forestry, Nanjing 211153, China; yunzhoulv@163.com (Y.-z.L.); hongguo417@163.com (X.-y.D.); huanglib@163.com (L.-b.H.); zjw932333@163.com (J.-w.Z.); hxd_519@163.com (X.-d.H.); sunhainan1989@hotmail.com (H.-n.S.)
\* Correspondence: jiangzeping518@126.com

**Abstract:** The Chinese elm is an important tree ecologically; however, little is known about its genetic diversity and adaptation mechanisms. In this study, a total of 107 individuals collected from seven natural populations in eastern China were investigated by specific locus amplified fragment sequencing (SLAF-seq). Based on the single nucleotide polymorphisms (SNPs) detected by SLAF-seq, genetic diversity and markers associated with climate variables were identified. All seven populations showed medium genetic diversity, with PIC values ranging from 0.2632 to 0.2761. AMOVA and Fst indicated that a low genetic differentiation existed among populations. Environmental association analyses with three climate variables (annual rainfall, annual average temperature, and altitude) resulted in, altogether, 43 and 30 putative adaptive loci by Bayenv2 and LFMM, respectively. Five adaptive genes were annotated, which were related to the functions of glycosylation, peroxisome synthesis, nucleic acid metabolism, energy metabolism, and signaling. This study was the first on the genetic diversity and local adaptation in Chinese elms, and the results will be helpful in future work on molecular breeding.

**Keywords:** Chinese elm; genetic diversity; adaptation; SLAF-seq; SNP loci

## 1. Introduction

Chinese elm (*Ulmus parvifolia*), which is native to China, Japan and Korea, has become a widely distributed ornamental tree that is frequently planted on lawns, along streets and in parks [1]. In China, the wild resources of *U. parvifolia* are mainly located in the northern and eastern areas, exhibiting a wide range of adaptation. Within this area, Chinese elm is recognized as a drought, heat, and cold tolerant tree [2–4]. Nevertheless, as global climate alteration will happen in the near future, it remains questionable to what degree the speed of future adaptation can keep up with the pace of climate change [5]. Therefore, an in-depth understanding of the genetic diversity and the genetic regulation of adaptation in Chinese elms is essential. Revealing polymorphisms and genes that determine adaptation would provide the basis for breeding genetically improved germplasms that could be used in changing environments.

Genetic diversity is the maximum of genetic variation presented in the genetic makeup of a specific species [6]. It is an important component of species biodiversity. Monitoring the genetic diversity of natural populations is of paramount importance, since it could shed light on the population structure, history, ecology, and adaptation of the species [7]. Local adaptation occurs gradually over time, with relatively long generation times. During the adaptation process, alleles that are best fitted to

the specific climate gradually prevail through positive selection [8]. Those alleles, once identified, can give new insights into plant adaptive evolution, as well as be utilized for future molecular breeding.

Previous research on genetic diversity and local adaptation of plants has been conducted at the DNA-based molecular level, such as simple sequence repeat (SSR), inter-simple sequence repeat (ISSR), random amplified polymorphic DNA (RAPD), amplified fragment length polymorphism (AFLP), and single nucleotide polymorphism (SNP) [7,9,10]. SNPs are genome sequence variations that occur when there is a single nucleotide change in the DNA sequence [11]. SNPs are the most abundant and stable type of DNA variation in a genome, therefore, the density of SNP markers is much higher than any other molecular markers [3]. Nowadays, reduced representation sequencing, such as genotyping-by-sequencing (GBS) and specific locus amplified fragment sequencing (SLAF-seq), has been used to quickly and efficiently identify numerous SNPs in plants [12,13]. As reduced representation sequencing can be performed without a reference genome, it has been tested on many kinds of trees, such as pecans [14], Japanese conifers [10], and *masson pine* [15].

To date, reports regarding the genetic diversity and adaptive mechanisms in Chinese elms remain remarkably scant. In this study, we attempt to explore the genetic diversity of seven natural populations of Chinese elms in eastern China, and then identify the potential local adaptation genes based on SLAF-seq identified SNPs. Our results might help in the marker-assisted breeding of Chinese elms in the future.

## 2. Materials and Methods

### 2.1. Plant Materials

Natural populations of *Ulmus parvifolia* were investigated in the present study. A total of seven populations with 107 individuals were collected from Jiangsu Province (XZ, JN, CS), Anhui Province (HUOS, HS), and Zhejiang Province (FY, LH). For each population, 13~17 individuals were sampled, with individuals at least 300 m apart. Collection details and climate information for the seven populations are summarized in Table 1 and Figure 1 Young healthy leaves were sampled and stored at −80 °C until further use.

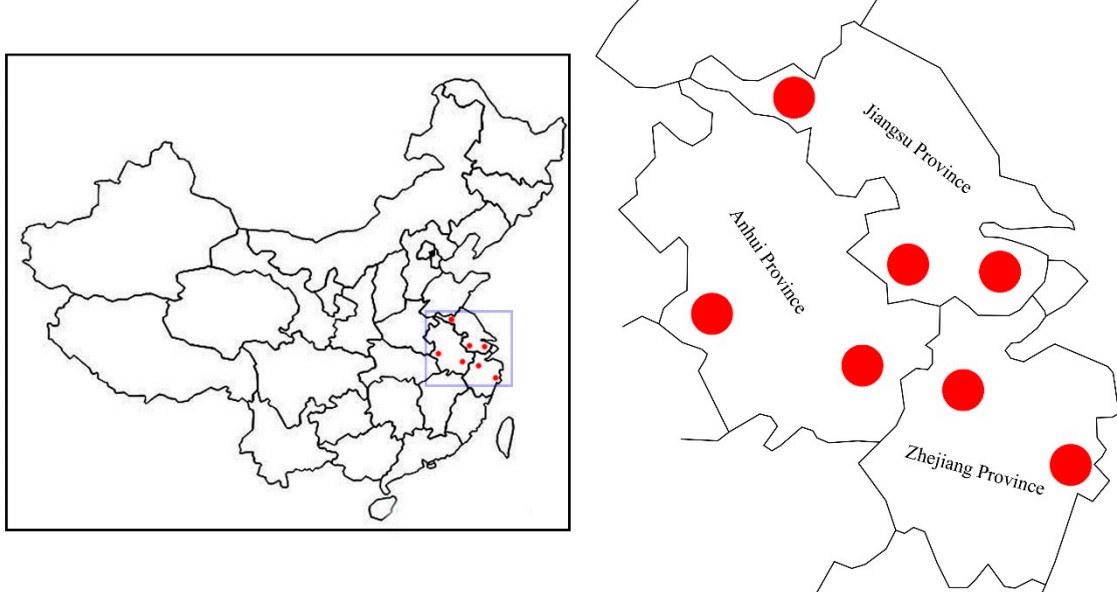

**Figure 1.** Map showing locations of the populations of Chinese elm.

**Table 1.** Population details of Chinese elms and their climate information.

| Population Location | Abbreviation | Sample Size | Annual Rainfall (cm) | Geographical Coordinates | Annual Average Temperature (°C) | Altitude (m) |
|---|---|---|---|---|---|---|
| Xuzhou, Jiangsu | XZ | 15 | 802.5 | 34°12′ N, 117°09′ E | 14.5 (0.7, 27.3) * | 56 |
| Jiangning, Jiangsu | JN | 13 | 1072.9 | 31°51′ N, 118°46′ E | 15.7 (2.9, 28.3) | 20 |
| Changsu, Jiangsu | CS | 17 | 1615.3 | 31°39′ N, 120°39′ E | 16.9 (3.6, 28.2) | 90 |
| Huoshan, Anhui | HUOS | 15 | 1366 | 31°26′ N, 116°23′ E | 15.3 (2.6, 27.7) | 110 |
| Huangshan, Anhui | HS | 14 | 2395 | 30°15′ N, 118°08′ E | 15.5 (4.4, 28.1) | 180 |
| Fuyang, Zhejiang | FY | 16 | 1441.9 | 30°03′ N, 119°37′ E | 16.1 (4.3, 28.8) | 90 |
| Linhai, Zhejiang | LH | 17 | 1550 | 28°47′ N, 121°34′ E | 17.1 (6.5, 28.6) | 10 |

* The values in brackets represent the average temperature over years in January and July.

## 2.2. High-Throughput Sequencing

About 20 mg of leaves were used for genomic DNA extraction via the DNeasy Plant Pro Kit (Qiagen, Hilden, Germany). DNA concentration and quality were assessed with a Nanodrop 1000 Spectrophotometer (Thermo Fisher Scientific, Wilmington, MA, USA) and 2% agarose gel electrophoresis. Quantified DNA samples were diluted to 100 ng/μL for the subsequent SLAF-seq analysis. SLAF-seq was performed according to a previous report [12], with some modifications. Since the genome of Chinese elm has not been published, we used the *Trema orientale* (the same species in Ulmaceae) for the prediction of enzyme digestion. Briefly, the reference genome of *Trema orientale* was used to perform marker discovery surveys through simulating in silico the number of markers obtained by various restriction enzymes. To get >100,000 SLAF tags that were evenly distributed in the genome, two restriction enzymes, *HinCII* and *HaeIII*, were finally selected. The efficiency of enzyme digestion was of importance for the reduced-representation sequencing. For the present study, *Oryza sativum ssp. japonica* DNA with a high-quality genomic information was used as a control to evaluate the quality of enzyme digestion. Following digestion, a single nucleotide (A) was added to the 3′ end using dATP at 37 °C, and then Dual-index adapters were ligated to the A-tailed DNA fragments. PCR amplification was subsequently performed using diluted restriction-ligation DNA as the template. The products of PCR were purified and pooled together. DNA fragments that were 414–464 in length were collected from agarose gel, and were chosen as SLAF tags. High-throughput sequencing was performed using an Illumina-HiSeq$^{TM}$ 2500 sequencing platform (Illumina, Inc.; San Diago, CA, USA) at Beijing Biomarker Technologies Corporation (Beijing, China).

## 2.3. SNP Calling

Raw reads generated from the sequencing platform were first qualified through removing the adapter sequence included in the raw reads, low-quality reads (quality scores < 20), and empty reads (reads just contained adapter sequence). High quality paired-end reads were clustered using the BLAT software based on sequence similarity [16]. Sequences with over 90% similarity among different individuals were identified as one SLAF locus [12]. Samtools [17] and the Genome Analysis Toolkit (GATK) [18] were used for SNP calling, and their intersection was considered to indicate reliable SNPs. For the phylogenetic analysis, SNPs with a minor allele frequency (MAF) < 5% and missing rate > 0.2 were filtered.

## 2.4. Diversity Analysis

A total of 457,888 SNPs from 107 individuals were developed to calculate the genetic diversity and population structure. The commonly used indexes of genetic diversity, including the observed allele number (Na), expected allele number (Ne), observed heterozygous number (Ho), expected heterozygous number (He), Nei's diversity index (H), Shannon's wiener index (I), and polymorphism information content (PIC), were calculated by POPGENE [19]. These indexes were calculated to estimate the degree of allele distribution (Na and Ne), genomic heterozygosity (Ho and He), gene diversity (H and I) and DNA polymorphism (PIC). In order to assess the population differentiation, Analysis of molecular variance (AMOVA) was calculated to estimate the partitioning of genetic variance among populations. Meanwhile, pairwise fixation index (Fst) among populations was also computed to detect how gene diversity was partitioned at each level. Inter-individual fixation index (FIS) was analyzed to determine the deviation of genotype frequencies from Hardy–Weinberg proportions within each population. AMOVA, Fst, and FIS were estimated by Arlequin [20].

The phylogenetic tree was constructed by MEGA X software with the following parameters: neighbor-joining method, Kimura 2-parameter model, and 1000 bootstrap replicates. The population structure was analyzed by Admixture [21], on the basis of the maximum-likelihood method. The number of populations (K), ranging from 1 to 10, was tested, and each individual was assigned to its respective populations according to the maximum membership probability.

### 2.5. Climatic Association Analysis

Two programs, Bayenv2 [22,23] and LFMM [24], were used to detect outlier loci that were possibly associated with climatic variables. First, we used Bayenv2 to detect correlations between SNP allele frequencies and environmental variables. A covariance matrix of allele frequencies was estimated across populations using the full set of SNPs to avoid population-specific effects. For each tested SNP, this program generated a Bayes factor (BF) and nonparametric Spearman's rank correlation coefficient ($\rho$) based on the Markov chain Monte Carlo (MCMC). In this study, the significance threshold for the putative adaptive makers were those ranked among the top 1% of BF values ($\log_{10}BF > 2.75$) and top 5% of $\rho$ values. The other software, LFMM, was also used for gene-climate association analysis. As it estimates the hidden impact of population structure, LFMM permits the presence of background levels of population structure (latent factors). The detected SNPs that exhibit an association with the environment were determined according to the $z$-score. Bonferroni adjustment was used on the $z$-score values for multiple tests. Markers with $z$-scores > 2.8 and a $p$-value < 0.01 were considered to be significant. Putative functions for the identified outlier loci were annotated using the NCBI and UniProt databases.

## 3. Results

### 3.1. SNP Detection

High-throughput sequencing based on SLAF generated a total of 439.74 M pair-end reads, with a mean GC content of 42.90%, and an average Q30 of 96.70%. We obtained a total of 2,059,418 high-quality SLAF tags for the 107 samples, with an average depth of 18.88x for each SLAF (Table 2). For the SLAF tags, 529,271 were polymorphic. These polymorphic SLAFs contained 4,138,972 SNPs in total, and 457,888 of them were utilized in further analysis after applying the filtering criteria.

**Table 2.** Summary of specific locus amplified fragment sequencing (SLAF-seq).

|  | No. of Reads | GC Content (%) | Q30 (%) | No. of SLAF | No. of Depth |
| --- | --- | --- | --- | --- | --- |
| Sum | 439,742,148 |  |  | 2,059,418 |  |
| Avg. | 4,109,739.70 | 42.9 | 96.7 | 19,246.90 | 18.88 |

### 3.2. Genetic Diversity and Genetic Differentiation

The value of the observed allele number (Na) was 2 across populations, and the values of the expected allele number (Ne) ranged from 1.5321 (XZ) to 1.5759 (FY), with a mean value of 1.5498. The observed heterozygous (Ho) values were significantly lower than the He values, with values lying between 0.1483 (CS) and 0.1822 (FY), and an average of 0.1599. The values of the expected heterozygous (He) number across the seven populations were between 0.3236 (XZ) and 0.3427 (FY), with an average value of 0.3315. Nei's diversity index (H) was within the range from 0.3385 (XZ) to 0.3570 (FY), with a mean value of 0.3467. Shannon's wiener index (I) varied from 0.4948 to 0.5171 for the XZ and FY populations, respectively. The PIC values of the seven populations ranged from 0.2632 to 0.2761, with an average of 0.2686. The maximum value of PIC was presented in the FY population, while the minimum value was found in the XZ population. As a measure of intragametophytic selfing, $F_{IS}$ were low in our study, varying from −0.03849 (FY population) to 0.06769 (HS population) (Table 3). All the $F_{IS}$ were on Hardy–Weinberg equilibrium ($p > 0.05$).

**Table 3.** Genetic diversity of seven Chinese elm populations.

| Population | Na | Ne | Ho | He | H | I | PIC | $F_{IS}$ |
|---|---|---|---|---|---|---|---|---|
| XZ | 2 | 1.5321 | 0.1582 | 0.3236 | 0.3385 | 0.4948 | 0.2632 | 0.04448 |
| JN | 2 | 1.5462 | 0.1572 | 0.3298 | 0.3476 | 0.5021 | 0.2675 | 0.05903 |
| CS | 2 | 1.5362 | 0.1483 | 0.3255 | 0.3390 | 0.4971 | 0.2645 | 0.05467 |
| HUOS | 2 | 1.5414 | 0.1510 | 0.3281 | 0.3435 | 0.5002 | 0.2664 | 0.06443 |
| HS | 2 | 1.5469 | 0.1523 | 0.3307 | 0.3475 | 0.5034 | 0.2682 | 0.06769 |
| FY | 2 | 1.5759 | 0.1822 | 0.3427 | 0.3570 | 0.5171 | 0.2761 | −0.03849 |
| LH | 2 | 1.5697 | 0.1699 | 0.3398 | 0.3534 | 0.5137 | 0.2741 | 0.01339 |

Na, observed allele number; Ne, expected allele number; Ho, observed heterozygous; He, expected heterozygous; H, Nei's diversity index; I, Shannon's wiener index; PIC, polymorphism information content; $F_{IS}$, inter-individual fixation index.

The pairwise fixation index (Fst) is a measure of genetic differentiation among populations. In our study, the lowest genetic differentiation existed between the HS and HUOS populations, with an Fst value of 0.00712. The LH and XZ populations presented the largest genetic differentiation, with an Fst value of 0.09106 (Table 4). AMOVA indicated that the maximum diversity occurred within individuals (92.22%), while the minimum diversity presented among individuals within populations (3.54%). A total of 4.24% of the genetic variation occurred among populations (Table 5).

**Table 4.** Pairwise fixation index (Fst) values among seven populations of Chinese elm.

| | XZ | JN | CS | HUOS | HS | FY |
|---|---|---|---|---|---|---|
| JN | 0.03396 | | | | | |
| CS | 0.03939 | 0.01504 | | | | |
| HUOS | 0.02869 | 0.01164 | 0.01525 | | | |
| HS | 0.04646 | 0.01845 | 0.017 | 0.00712 | | |
| FY | 0.08368 | 0.05256 | 0.04706 | 0.04659 | 0.04891 | |
| LH | 0.09106 | 0.0536 | 0.04831 | 0.05281 | 0.05296 | 0.07493 |

**Table 5.** Analysis of molecular variance (AMOVA) of genetic diversity of Chinese elm populations.

| Source of Variation | df | Sum of Squares | Variance Components | Percentage of Variation (%) |
|---|---|---|---|---|
| Among populations | 6 | 30,354.88 | 93.79375 | 4.24 |
| Among individuals within populations | 100 | 219,566.2 | 78.17923 | 3.54 |
| Within individuals | 107 | 218,205.5 | 2039.304 | 92.22 |

*3.3. Phylogenetic Relationship and Population Structure*

The genetic relationships of the 107 individuals were exemplified by a phylogenetic tree. Interestingly, we found that the individuals could not be divided into distinct clades, which indicates a weak population structure of the individuals (Figure 2). Generally, individuals in the same subclade were from the same population (Figure 2).

The genetic structure of the *U. parvifolia* populations was assessed with the Admixture software. As shown in Figure 3, the lowest K-values were detected when K = 1, indicating that a weak population structure existed in the individuals. A relatively low K-values was seen when K = 2, and correspondingly, the 107 individuals could be categorized into two groups. Group I contained 91 individuals, which were mainly from the FY, MH, HS, XZ, JN, and CS populations. Group II consisted only of 16 individuals from the LH population (Figure 4). Individuals with a low degree of admixture were seen from all the studied populations.

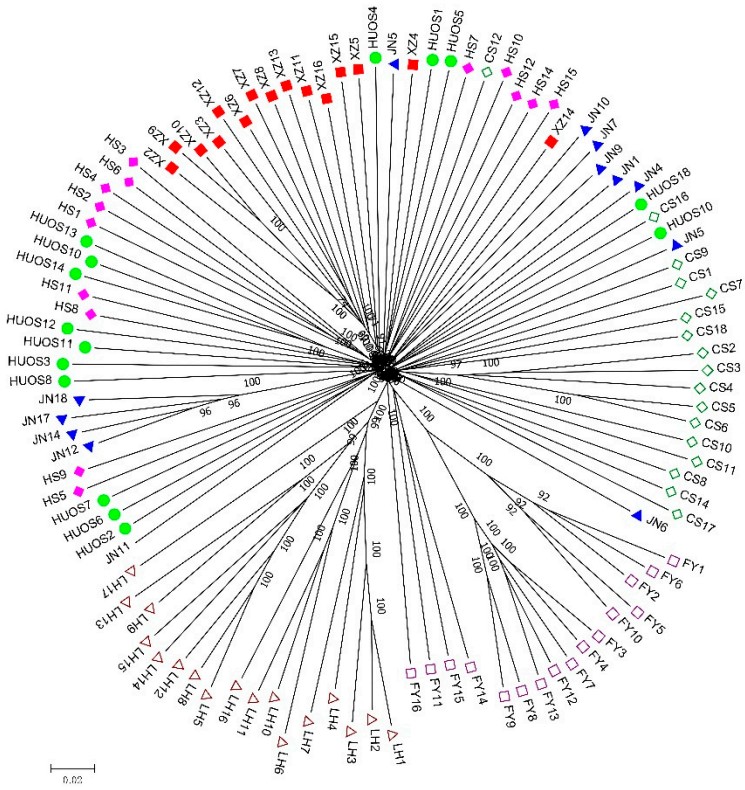

**Figure 2.** Phylogenetic tree of the 197 individuals based on the analysis of 457,888 single nucleotide polymorphisms (SNPs).

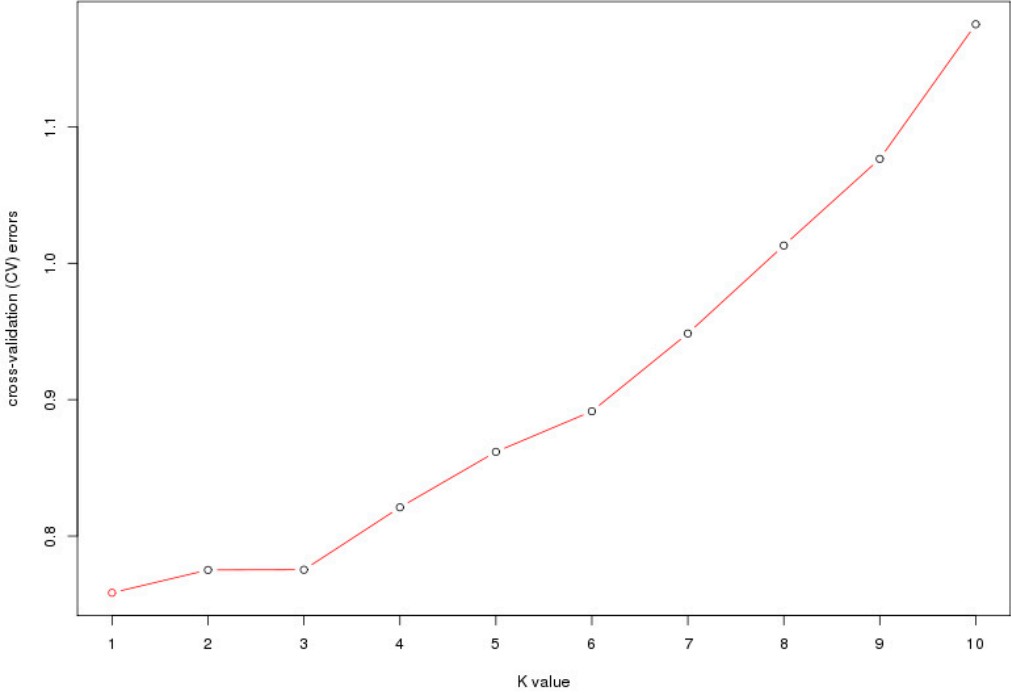

**Figure 3.** ADMIXTURE estimation of the number of groups for K values ranging from 1 to 10.

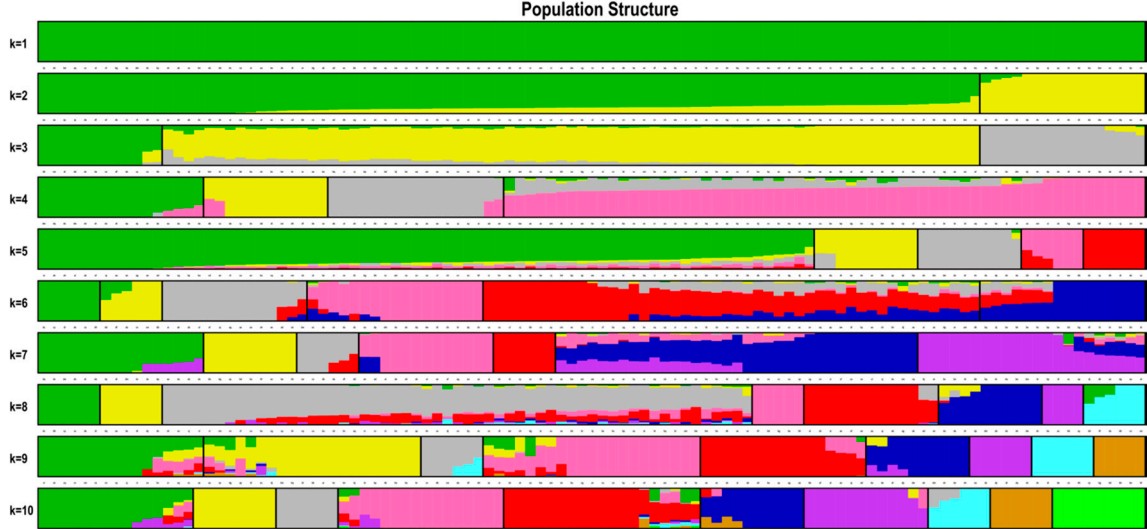

**Figure 4.** Population structure analysis of the 107 individuals based on 457,888 SNPs. The bars in the x-axis indicate different individuals. Colors in each row represent structural components.

*3.4. Association between SNP Markers and Environmental Variables*

The association analysis of SNPs and environmental variables was conducted by the Bayenv2 and LFMM programs. Bayenv2 analysis identified a total of 43 SNP markers showing significant correlation with the environmental variables. Of these, 8, 10, and 25 markers were associated with altitude, annual rainfall, and annual average temperature, respectively (Table 6). A set of 30 markers associated with climatic variables was obtained by the LFMM program. The highest number of associations was for temperature, which was related to 16 markers; the annual rainfall and altitude were correlated with 4 and 10 markers, respectively (Table 7). Blast searches indicated that five of the correlated SNP markers could be annotated. Two markers (Marker204041 and Marker76627) associated with altitude could be annotated to the *DEAD-box helicase* and *V-type proton ATPase* genes, respectively. The SNP markers, Marker68303 and Marker129188, correlated with annual rainfall, were found in the regions of the *UDP-glycosyltransferase* (*UGT*) and *peroxisome biogenesis protein* genes, respectively. The SNP marker, Marker87380, associated with annual average temperature, seems to underlie the *Cysteine-rich receptor-like protein kinase* gene (Table 8).

**Table 6.** A summary of putative adaptive markers displaying associations with different climate variables identified by Bayenv2 analysis.

| SNP ID | Pos | log10 (BF) | $\rho$ | Altitude | Annual Rainfall | Annual Average Temperature |
|--------|-----|-----------|--------|----------|-----------------|----------------------------|
| Marker127061 | 164 | 4.9954 | 0.1019 | * | | |
| Marker58279 | 222 | 4.1913 | 0.1049 | * | | |
| Marker204041 | 6 | 4.1152 | 0.1288 | * | | |
| Marker971355 | 147 | 3.8946 | 0.1053 | * | | |
| Marker62103 | 215 | 3.5162 | 0.1594 | * | | |
| Marker76627 | 243 | 3.3182 | 0.1087 | * | | |
| Marker54757 | 5 | 2.3134 | 0.1045 | * | | |
| Marker33355 | 188 | 2.0794 | 0.1071 | * | | |
| Marker129188 | 112 | 25.6440 | 0.1808 | | * | |
| Marker39336 | 258 | 11.2230 | 0.1233 | | * | |
| Marker201123 | 242 | 7.7245 | 0.1258 | | * | |
| Marker44387 | 8 | 6.1236 | 0.1641 | | * | |
| Marker40822 | 186 | 3.5858 | 0.1087 | | * | |
| Marker68303 | 61 | 2.9905 | 0.1092 | | * | |
| Marker85734 | 99 | 2.8768 | 0.1019 | | * | |

**Table 6.** *Cont.*

| SNP ID | Pos | log10 (BF) | ρ | Altitude | Annual Rainfall | Annual Average Temperature |
|---|---|---|---|---|---|---|
| Marker71488 | 44 | 2.8301 | 0.1134 | | * | |
| Marker37050 | 59 | 2.6988 | 0.1115 | | * | |
| Marker58475 | 77 | 2.2600 | 0.1380 | | * | |
| Marker41305 | 13 | 37.3380 | 0.1019 | | | * |
| Marker62540 | 103 | 23.1980 | 0.1043 | | | * |
| Marker107170 | 225 | 22.4260 | 0.1838 | | | * |
| Marker65958 | 170 | 21.5660 | 0.1102 | | | * |
| Marker18102 | 240 | 20.3570 | 0.1169 | | | * |
| Marker18740 | 56 | 13.0760 | 0.1192 | | | * |
| Marker60000 | 154 | 11.8650 | 0.1013 | | | * |
| Marker78616 | 181 | 11.4110 | 0.1134 | | | * |
| Marker29379 | 156 | 10.9290 | 0.1295 | | | * |
| Marker112072 | 227 | 7.3602 | 0.1023 | | | * |
| Marker62404 | 182 | 7.2272 | 0.1218 | | | * |
| Marker116716 | 258 | 6.6439 | 0.1066 | | | * |
| Marker211065 | 76 | 4.9942 | 0.1024 | | | * |
| Marker145731 | 238 | 4.6537 | 0.1030 | | | * |
| Marker2846306 | 228 | 4.0441 | 0.1947 | | | * |
| Marker32405 | 256 | 3.6509 | 0.1019 | | | * |
| Marker109530 | 179 | 3.5689 | 0.1013 | | | * |
| Marker24830 | 245 | 3.4366 | 0.1058 | | | * |
| Marker60529 | 214 | 3.4104 | 0.1613 | | | * |
| Marker29189 | 49 | 3.0557 | 0.1021 | | | * |
| Marker51336 | 148 | 2.8289 | 0.1259 | | | * |
| Marker31333 | 33 | 2.4288 | 0.1735 | | | * |
| Marker130465 | 24 | 2.3988 | 0.1026 | | | * |
| Marker65370 | 245 | 2.3645 | 0.1109 | | | * |
| Marker41605 | 187 | 2.0949 | 0.1111 | | | * |

\* suggests that the SNP showed an association with that specific climate variable.

**Table 7.** A summary of putative adaptive markers displaying associations with different climate variables identified by LFMM analysis.

| SNP ID | Position | Z-Scores | log10(p) | *p*-Value | Altitude | Annual Rainfall | Annual Average Temperature |
|---|---|---|---|---|---|---|---|
| Marker45074 | 257 | 2.94 | 2.39 | 0.0041 | * | | |
| Marker172745 | 21 | 2.90 | 2.34 | 0.0046 | * | | |
| Marker45074 | 62 | 2.88 | 2.32 | 0.0048 | * | | |
| Marker45074 | 10 | 2.87 | 2.30 | 0.0050 | * | | |
| Marker45074 | 12 | 2.86 | 2.29 | 0.0051 | * | | |
| Marker45074 | 69 | 2.86 | 2.29 | 0.0051 | * | | |
| Marker45074 | 77 | 2.85 | 2.28 | 0.0052 | * | | |
| Marker45074 | 253 | 2.85 | 2.28 | 0.0053 | * | | |
| Marker45074 | 197 | 2.85 | 2.27 | 0.0053 | * | | |
| Marker141529 | 57 | −2.82 | 2.24 | 0.0057 | * | | |
| Marker101622 | 111 | 2.93 | 2.38 | 0.0042 | | * | |
| Marker101622 | 23 | 2.92 | 2.37 | 0.0043 | | * | |
| Marker101622 | 75 | 2.92 | 2.37 | 0.0043 | | * | |
| Marker101622 | 74 | 2.80 | 2.22 | 0.0061 | | * | |
| Marker147012 | 258 | −2.95 | 2.40 | 0.0040 | | | * |
| Marker147012 | 172 | 2.94 | 2.40 | 0.0040 | | | * |
| Marker147012 | 147 | 2.94 | 2.40 | 0.0040 | | | * |
| Marker147012 | 59 | −2.94 | 2.40 | 0.0040 | | | * |
| Marker79339 | 199 | 2.94 | 2.40 | 0.0040 | | | * |

**Table 7.** *Cont.*

| SNP ID | Position | Z-Scores | log10(p) | *p*-Value | Altitude | Annual Rainfall | Annual Average Temperature |
|---|---|---|---|---|---|---|---|
| Marker112495 | 159 | −2.94 | 2.40 | 0.0040 | | | * |
| Marker112495 | 155 | 2.94 | 2.40 | 0.0040 | | | * |
| Marker79339 | 63 | 2.94 | 2.40 | 0.0040 | | | * |
| Marker87380 | 208 | 2.94 | 2.40 | 0.0040 | | | * |
| Marker102725 | 233 | −2.94 | 2.39 | 0.0040 | | | * |
| Marker43598 | 85 | 2.94 | 2.39 | 0.0040 | | | * |
| Marker58232 | 213 | 2.94 | 2.39 | 0.0041 | | | * |
| Marker106952 | 19 | −2.94 | 2.39 | 0.0041 | | | * |
| Marker112495 | 237 | −2.94 | 2.39 | 0.0041 | | | * |
| Marker106952 | 94 | −2.94 | 2.39 | 0.0041 | | | * |
| Marker112495 | 238 | −2.94 | 2.39 | 0.0041 | | | * |

* suggests that the SNP showed an association with that specific climate variable.

**Table 8.** Identification of putative candidate genes of the associated SNP markers.

| Climate Variables | Marker ID | Position | Putative Genes |
|---|---|---|---|
| Altitude | Marker204041 | 6 | DEAD-box helicase |
| | Marker76627 | 243 | V-type proton ATPase |
| Annual rainfall | Marker68303 | 61 | UDP-glycosyltransferase |
| | Marker129188 | 112 | Peroxisome biogenesis protein |
| Annual average temperature | Marker87380 | 208 | Cysteine-rich receptor-like protein kinase |

## 4. Discussion

The present study is the first attempt to use SNPs derived from SLAF to assess the genetic diversity and explore the adaptation mechanisms of Chinese elms. In recent years, SLAF-seq technology has become a low-cost technique to effectively develop reliable SNP and InDel markers for genome-wide association analysis and high-density genetic map construction [25,26]. Our study identified a total of 4,138,972 SNPs and selected 457,888 SNPs with MAF > 5% and a missing rate < 0.2 for further analysis. The number of molecular markers was dramatically larger than that in previous reports on elm species [27,28], which facilitates precise genetic analysis.

Heterozygosity is an important measure of overall genetic diversity [25]. In our study, the Ho and He values ranged from 0.1483 to 0.1822 (an average of 0.1599) and 0.3236 to 0.3427 (an average of 0.3315), respectively (Table 3). These values were lower than the results observed in other trees [25,29]. A relative lower level of genetic heterozygosity for the Chinese elms might be due to the existence of spatial isolation in different groups, hindering the gene communication between individuals to some extent. The index, PIC, measures the degree of informativeness of a genetic marker, with values ranging from 0 to 1 [29]. A locus with a PIC value of 0 is undesirable [30]. When PIC < 0.25, it indicates a low polymorphism, and 0.25 < PIC < 0.50 represents a median polymorphism. In contrast, PIC > 0.50 is indicative of high polymorphism [31]. According to this criteria, as the PIC values were between 0.2632 to 0.2761 (Table 3), the tested seven populations in our study possessed medium genetic diversity in terms of PIC. The inter-individual fixation index (FIS) measures the deviation of genotype frequencies from Hardy–Weinberg proportions within each population. A negative FIS indicates heterozygote excess (outbreeding), while a positive value reflects a deficiency in heterozygosity (inbreeding) [32]. In our study, the FY population presented a negative FIS (−0.03849) (Table 3), suggesting a slight excess of heterozygotes. The other six populations exhibited a positive FIS (Table 3). All the populations were not statistically significantly deviated from the Hardy–Weinberg equilibrium ($p > 0.05$), indicating a relatively random mating for these populations. Overall, the FY population displayed higher genetic diversity than the other six populations, which was supported by the larger Ne, Ho, He, H, I, and PIC values (Table 3).

Outcrossing woody plants tend to possess low levels of genetic differentiation among populations [33]. In the current study, differentiation among populations was estimated by Fst values. Fst > 0.25 signifies a great genetic differentiation, 0.25 > Fst > 0.15 indicates a moderate genetic differentiation, 0.15 > Fst > 0.05 means a small genetic differentiation, and Fst < 0.05 represents negligible genetic differentiation [34]. Based on this standard, a low genetic differentiation was found among the studied populations (Fst values ranging from 0.00712 to 0.09106) (Table 4). Additionally, AMOVA analysis (Table 5) also indicated a low percentage of variation (4.24%) among populations. Similar results could be found in other trees [7,35].

Investigating the population structure of tested individuals is the premise for association analysis, since the presence of the population structure could affect the validity of association results [36–38]. In our study, the optimal K value of the seven populations was 1 (Figure 2), indicating no population structure existed in the studied groups. The geographic boundaries had a weak effect on the genetic structure of Chinese elms. The existence of population structure might cause correlations between unlinked locis, and would usually result in increased false associations, the weak population structure of Chinese elm in our study would be conducive to subsequent association analysis.

Natural selection has an important impact on shaping the genetic variation of a population, and therefore promotes local adaptation [39]. In this research, based on the identified SNP markers, an association study was used to uncover the hidden genetic basis of local adaptation. The associated SNP markers were blasted against public databases for putative genes. We found that the genes of *DEAD-box helicase* and *V-type proton ATPase* seemed to be candidates for adaptation to altitude. *DEAD-box helicase* is involved in nucleic acid metabolism functions, such as transcription, translation, replication, repair, recombination, ribosome biogenesis and splicing, which control plant grow and development [40]. *V-type ATPase*, as a transporter, is essential for energy metabolism and maintenance of solute homeostasis, which makes it indispensable for plant growth [41,42]. *V-type proton ATPase* has been shown to play a significant role in plant adaptation to stressful growth conditions [42]. We deduced that variations in altitude would lead to a difference in plant growth according to the functions of *DEAD-box helicase* and *V-type proton ATPase*.

*UDP-glycosyltransferase* (*UGT*) and *peroxisome biogenesis protein* were associated with annual rainfall variable. *UGT* belongs to the *glycosyltransferase* (*GT*) multigene family [43]. In plants, GTs are a ubiquitous group of enzymes involved in the glycosylation process, and glycosylation leads to the formation of glycosylated secondary chemicals such as flavonols, anthocyanins, and plant hormones [44,45]. Glycosylated secondary products possess increased water solubility and molecule stability, which could change their biological activity [44]. *Peroxisome biogenesis protein* might participate in the synthesis of peroxisomes, a metabolic organelle that exists in all eukaryotic cells [46]. Peroxisomes contribute to resistance against oxidative stresses, β- and α-oxidation of fatty acids, and synthesis of ether lipids [47,48]. The products of *UGT* and *Peroxisome biogenesis protein* seemed to confer advantages for plants survival in rainy climate [43]. It is reasonable that *UGT* and *peroxisome biogenesis protein* appeared as candidates for adaptation to rainfall climate.

*Cysteine-rich receptor-like protein kinase* (*CRK*) was the putative gene that we found was associated with the annual average temperature variable. CRKs are critical signaling components that regulate plant developmental and defense processes. In *Arabidopsis*, overexpression of a CRK gene confers drought tolerance without affecting plant growth [49]. Considering that the temperature variable would be generally correlated with drought stress, it is possible that there may be a difference in drought-associated loci among populations. Identification of putative candidate genes correlated with the environment would reveal a primary insight into functional genes mediating local adaptation. However, further studies are required in the future to explain the accurate roles of those candidate genes in the adaptation processes of Chinese elms.

## 5. Conclusions

The present study analyzed the genetic diversity and adaptation of seven natural populations of Chinese elms in eastern China. The trees were genotyped by SLAF-seq technology, and then identification of SNPs was carried out. The natural population of Chinese elms showed a moderate level of genetic diversity (PIC = 0.2632~0.2761), low level of genetic differentiation, and a simple population structure (K = 1). The association analysis of genetic markers and environmental factors resulted in putative markers involved in local adaptation. A blast search was conducted to detect underlying putative candidate genes for the correlated markers. A total of five genes could be annotated, which were related to the functions of glycosylation, peroxisome synthesis, nucleic acid metabolism, energy metabolism, and signaling. The results will be helpful for future work on molecular breeding of this species.

**Author Contributions:** Y.-z.L. carried out the experiments, data analyses, drafted the manuscript and participated in the project design; Z.-p.J. chiefly designed the project, supervised the research and reviewed the manuscript; X.-y.D., J.-w.Z. and H.-n.S. collected the phenotypic data, L.-b.H. and X.-d.H. participated in the project design and data analyses. All authors have read and agreed to the published version of the manuscript.

**Funding:** This research was funded by the Modern Agricultural Projects by the Agriculture Department of Jiangsu Province, grant number BE2017386; Jiangsu Agriculture Science and Technology Innovation Found (JASTIF), grant number CX(17)2026; the National Natural Science Foundation of Jiangsu Province, grant number BK20141041. And the APC was funded by BE2017386.

**Conflicts of Interest:** The authors declare no conflict of interest.

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
