# Peer review of "SLAF-seq Uncovers the Genetic Diversity and Adaptation of Chinese Elm (Ulmus parvifolia) in Eastern China"

_forests, doi:10.3390/f11010080_

Round 1

Reviewer 1 Report

This manuscript samples seven populations of an important ornamental tree, Chinese elm, in order understand population structure and adaptation of the species. The paper is well written, but I think overall lacks citations. I have highlighted particular places citations are missing in the line comments below. More robust citations on population genetics and local adaptation could be added to the introduction, many of the methods lack citations, and the discussion could include more citations and comparisons of this study to them for broader implications and applications of this study. I am not an expert on the methods used and I suspect many readers will not be as well. I have suggested that more explanation should be given about the diversity analyses used in the methods as well.

Lines 29-30: Are there any citations you could possibly add for the sentence beginning, “Within this area, Chinese elm…”

Line 36-37: If possible, please provide a better definition of genetic diversity with a citation. I believe this one is from Wikipedia.

Line 48-49: Please add a citation.

Line 54-55: Be consistent with using either the common names or the Latin binomial for these species. Maybe change Pinus massoniana to its common name.

Line 63: How far apart were populations? Could you maybe provide a map as a supplement? Since you found K=1, it would be good to know the distance between populations.

Line 66-67: I would reword this sentence for clarity. Maybe, “From each population, 13-17 individuals were sampled, with individuals at least 300 m apart.”

Line 69: All are of the forests these populations are found in primary forests? Are there major differences between the forests that could affect genetic diversity?

Line 97: More detail could be given about each diversity analysis, like what Nei’s diversity index is/means.

Line 102: Add citation for POPGENE

Line 108: change “population” to “populations”

Line 112: Add citations for Bayenv2 and LFMM

Line 122: change “will be” to “were”

Table 3: Fis is not included in the footer of the table.

Line 162-165: I find the explanation for the groupings of populations not consistent with Figure 1. For instance, Group III contains the majority of individuals from HUOS and HS, but they are not listed and oppose the statement on Line 165 that “Generally, individuals from the same populations were grouped together.” This statement only seems consistent for populations LH, FY, and XZ, with all the other populations having individuals spread between multiple groups. How were these three groups chosen? It may be better to just present the phylogeny without grouping them, especially since the following results suggest at most a K-value of 2.

Line 188: Change “two” to “Two”

Table 8: Add what “Pos” means to the table information.

Line 212-214: Please add citations for the sentence beginning, “A relative lower level of genetic…”

Discussion: please reference back to the tables that show these results, for example Line 219 include a reference for Table 3 that shows the PIC values.

Line 242-243: Add a citation and explain why weak population structure is better for association analyses.

Lines 257-266: Could you relate the functions of these genes back to rainfall variation more in this paragraph.

Reviewer 2 Report

This is an interesting work in which a deep sequencing method (SLAF-seq) was applied to describe the genetic diversity of Chinese elms (Ulmus parvifolia) in Eastern China, based on SNP markers. Furthermore, association analyses with three climate variables revealed putative adaptive genes.

Although the results are well-presented, there are certain weak points that should be addressed.

Lns. 36-37. The definition of genetic diversity raises questions. Is there any reference to cite?

Lns. 64-66. Some explanation for the reason why these particular populations were selected is missing.

Ln. 71. Do you mean “values in brackets"?

Lns. 78-80. Why the genome of Trema orientale was used? It needs explanation, since it is phylogenetically distant from Ulmus parvifolia.
Lns. 80-81 (& lns. 98-99). What was the expected 'sequencing coverage'?

Lns.  81-82. I cannot understand the reason why Oryza sativum was used instead of Ulmus parvifolia.

Ln. 83. How  a single nucleotide (A) was added to the 3 ?

Ln. 91 Please explain “empty reads”.

Ln. 93. Reference?

Round 2

Reviewer 1 Report

This manuscript samples seven populations of an important ornamental tree, Chinese elm, in order understand population structure and adaptation of the species. This manuscript is timely as researchers are often searching for genetic tools that are inexpensive and allow for determination of local adaptation without a genome. The authors have addressed all of my previous comments by including more detail in the methods and citations throughout the manuscript. I have only a couple of minor comments included below:

Line 32: insert “of” between “understanding” and “the”

Lines 80-81: I do not understand what “(the same species in Ulmaceae)” means. Trema is in the family Cannabaceae.

Line 108: Change “Those indexes” to “These indexes”

Line 258: Deleted “an” before “increased”

Line 282: Change “plants survived” to “plant survival”

Line 281-282: Please add a citation for the sentence starting, “The products of UGT and Peroxisome…”

Line 283:delete “were” before “appeared”
